# ELKS controls the pool of readily releasable vesicles at excitatory synapses through its N-terminal coiled-coil domains

Richard G Held, Changliang Liu, Pascal S Kaeser*

Department of Neurobiology, Harvard Medical School, Boston, United States

**Abstract** In a presynaptic nerve terminal, synaptic strength is determined by the pool of readily releasable vesicles (RRP) and the probability of release (P) of each RRP vesicle. These parameters are controlled at the active zone and vary across synapses, but how such synapse specific control is achieved is not understood. ELKS proteins are enriched at vertebrate active zones and enhance P at inhibitory hippocampal synapses, but ELKS functions at excitatory synapses are not known. Studying conditional knockout mice for ELKS, we find that ELKS enhances the RRP at excitatory synapses without affecting P. Surprisingly, ELKS C-terminal sequences, which interact with RIM, are dispensable for RRP enhancement. Instead, the N-terminal ELKS coiled-coil domains that bind to Liprin-$\alpha$ and Bassoon are necessary to control RRP. Thus, ELKS removal has differential, synapse-specific effects on RRP and P, and our findings establish important roles for ELKS N-terminal domains in synaptic vesicle priming.

## Introduction

Within a presynaptic nerve terminal, synaptic vesicle exocytosis is restricted to sites of neurotransmitter release called active zones. The active zone is a dense protein complex that is attached to the presynaptic plasma membrane and is exactly opposed to postsynaptic receptors (*Couteaux and Pecot-Dechavassine, 1970*; *Schoch and Gundelfinger, 2006*; *Südhof, 2012*). At the active zone, a small subset of the synaptic vesicles are primed in close proximity to presynaptic $Ca^{2+}$ channels such that the incoming action potential leads to neurotransmitter release with minimal delay. The proteins of the active zone control the size of this pool of primed, readily releasable vesicles (RRP) and the release probability of those vesicles in response to an action potential (*Kaeser and Regehr, 2014*; *Alabi and Tsien, 2012*). Vesicular release probability, referred to in this paper as P, and RRP size together act to set synaptic strength, sometimes referred to as the synaptic probability of release (*Stevens, 2003*; *Zucker and Regehr, 2002*). It is well known that RRP size and vesicular release probability differ across synapses, contributing to the generation of unique release properties (*Abbott and Regehr, 2004*). In the hippocampus, for example, excitatory and inhibitory synapses have markedly different properties (*Kraushaar and Jonas, 2000*; *Salin et al., 1996*). The underlying molecular mechanisms that control RRP and P are still only partially understood, and it is not known what components of the release machinery account for their synapse specific control.

ELKS, RIM, Munc13, RIM-binding protein (RIM-BP), Bassoon/Piccolo, and Liprin-$\alpha$ proteins form a protein complex that defines the active zone (*Schoch and Gundelfinger, 2006*; *Südhof, 2012*). This protein complex includes many additional proteins that are not active zone specific (*Boyken et al., 2013*; *Muller et al., 2010*; *Schoch and Gundelfinger, 2006*). ELKS (also called Erc, CAST, and Rab6IP2) was identified as an active zone protein through its interactions with RIM and named for its high content in the amino acids E, L, K, and S (*Nakata et al., 1999*; *Wang et al., 2002*; *Monier et al., 2002*; *Ohtsuka et al., 2002*). ELKS has known in vitro interactions with many active

*For correspondence: kaeser@hms.harvard.edu

**eLife digest** Nerve cells in the brain communicate with one another at connections known as synapses: one nerve cell releases signaling molecules called neurotransmitters into the synapse, which are then sensed by the second cell. For the brain to work correctly, it is important that the nerve cells control when and how much neurotransmitter they release. Nerve cells package neurotransmitters into small packets called vesicles. These vesicles can be released at the so-called active zones of each synapse, though only a small subset of vesicles at a synapse are releasable.

Many proteins at the active zone control the release of vesicles to influence how nerve cells communicate with each another. ELKS is one of the proteins found at the active zones of nerve cells that release either of the two most common neurotransmitters in the brain: glutamate and GABA. Held et al. have now found that the ELKS protein affects the release of these two neurotransmitters in different ways in the two types of nerve cells. The experiments showed that the number of releasable neurotransmitter-filled vesicles was lower in mouse nerve cells that release glutamate when the genes for the ELKS proteins were deleted in these cells. When the ELKS genes were deleted in the nerve cells that release GABA, the number of releasable vesicles remained the same, though the vesicles were less likely to be released.

The fact that removing ELKS has different effects at these two types of synapses suggests that the active zone is not the same at all synapses. Furthermore, these results imply that ELKS is capable of fine-tuning the communication between nerve cells. Future experiments will address how glutamate- and GABA-releasing active zones differ at the molecular and structural levels. Ultimately, this will lead to a better understanding of how information is processed in the brain.

zone proteins and additional neuronal proteins (*Figure 1A*). It contains a C-terminal region that binds to the PDZ domain of RIM (*Wang et al., 2002*; *Ohtsuka et al., 2002*) and multiple coiled-coil stretches, which we have subdivided based on homology between the various vertebrate and invertebrate ELKS isoforms into four coiled-coil domains (CC$_A$-CC$_D$, *Figure 1A*). The coiled-coil stretches bind in vitro to Liprin-α (CC$_A$-CC$_C$, [*Ko et al., 2003*]), Bassoon (CC$_C$, [*Takao-Rikitsu et al., 2004*]), and β-subunits of Ca$^{2+}$ channels (CC$_D$, [*Kiyonaka et al., 2012*]). Vertebrate genomes contain two genes for ELKS, *Erc1* and *Erc2* (*Wang et al., 2002*), whereas *C.elegans* expresses a single ELKS homolog (*Deken et al., 2005*). *D. melanogaster* expresses a protein called Brp with homology to ELKS in the N-terminal but not the C-terminal half (*Wagh et al., 2006*; *Kittel et al., 2006*; *Monier et al., 2002*). Vertebrate ELKS proteins are expressed as predominant, synaptic α-isoforms and shorter β-variants, which account for less than 5% of ELKS (*Kaeser et al., 2009*; *Liu et al., 2014*). In addition, ELKS C-terminal variants determine RIM-binding: the B-isoforms are prominently expressed in the brain and contain the RIM binding site, whereas A-isoforms are expressed outside the brain and lack RIM binding (*Wang et al., 2002*; *Kaeser et al., 2009*).

The observation that ELKS binds to several active zone proteins has led to the hypothesis that ELKS scaffolds other active zone proteins, in particular RIM (*Takao-Rikitsu et al., 2004*; *Ohtsuka, 2013*; *Ohtsuka et al., 2002*). Invertebrate studies offer mixed support to this hypothesis. Loss of Brp disrupts the T-bar structures at the fly neuromuscular junction (*Kittel et al., 2006*), but this function involves the C-terminal region of Brp (*Fouquet et al., 2009*). In contrast, *C. elegans* ELKS is not required for recruitment of other active zone proteins (*Deken et al., 2005*), but a gain of function mutation in syd-2, the *C. elegans* Liprin-α homologue, requires ELKS for its synaptogenic activity (*Dai et al., 2006*).

Relatively little is known about the role and molecular mechanisms of ELKS in neurotransmitter release at vertebrate synapses. Previous studies showed that ELKS1α/2α boost Ca$^{2+}$ influx at inhibitory synapses, whereas ELKS2α has a regulatory, non-essential function in RRP at these synapses (*Liu et al., 2014*; *Kaeser et al., 2009*). In contrast, at excitatory ribbon synapses of rod photoreceptors, ELKS2α/CAST may have a structural role to enhance synaptic transmission (*tom Dieck et al., 2012*). These studies suggest the interesting possibility that ELKS may have differential roles in synaptic transmission between synapses. Thus far, ELKS functions have not been studied at small, excitatory synapses, the most abundant synapses in the vertebrate brain. Here, we establish that ELKS is

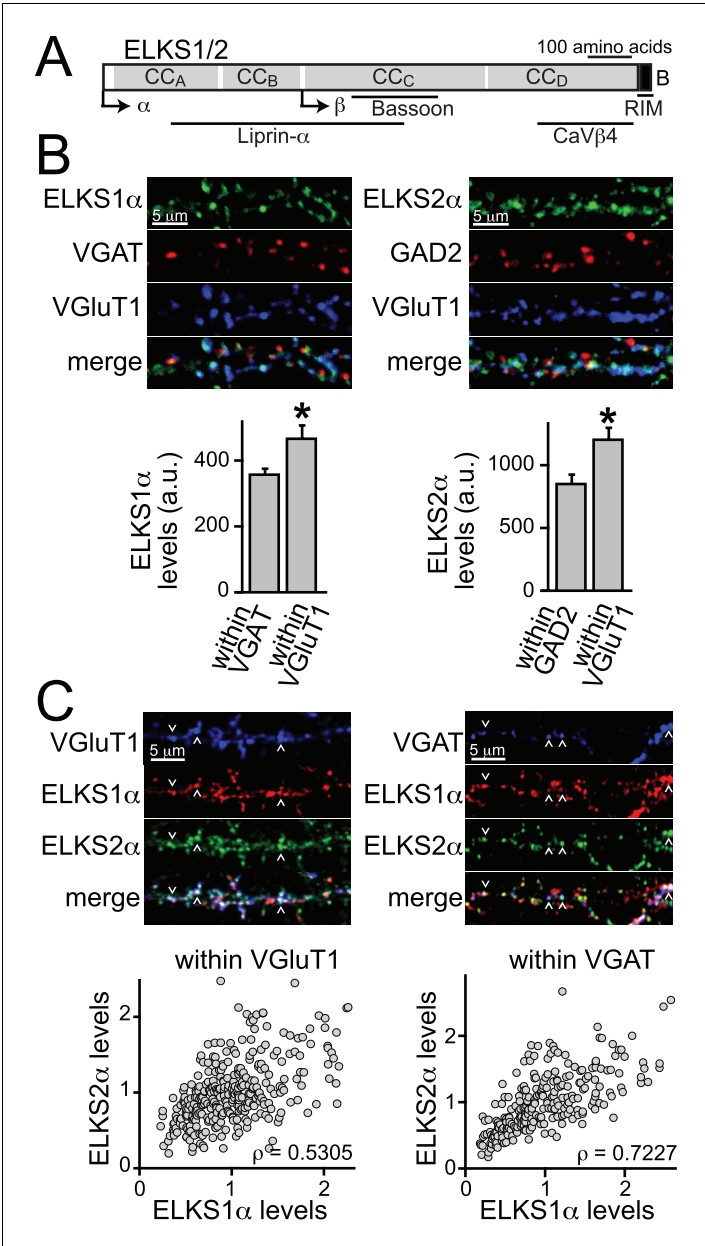

**Figure 1.** ELKS1α and ELKS2α are co-expressed at excitatory synapses. (**A**) Schematic of ELKS protein structure. Arrows: transcriptional start sites of α- and β-ELKS, $CC_{A-D}$: coiled-coil regions A - D (ELKS1: $CC_A$ [1]MYG...SKI[208], $CC_B$ [209]TIW...ENN[358], $CC_C$ [359]MLR...EAT[696], $CC_D$[697]LEA...EEE[988]; ELKS2: $CC_A$[1]MYG...ARM[204], $CC_B$[205]SVL...ENI[362], $CC_C$[363]HLR...NIE[656], $CC_D$[657]DDS...DEE[917], B: PDZ-binding sequence (ELKS1: [989]GIWA[992], ELKS2: [918]GIWA[921]) of the ELKS-B C-terminal splice variant. Binding regions for interacting active zone proteins are indicated with black bars. (**B**) Sample images and quantification of ELKS1α (left) and ELKS2α (right) expression levels at excitatory and inhibitory synapses. VGAT or GAD2 (red, inhibitory synapses) and VGluT1 (blue, excitatory synapses) staining was used to define regions of interest (ROIs), respectively (control n = 4 independent cultures, cDKO n = 4, 10 images were averaged per culture). All data are means ± SEM; *$p \leq 0.05$ as determined by Student's t test. (**C**) Sample images (top) and correlation of expression levels of ELKS1α and ELKS2α (bottom) at excitatory (left) and inhibitory (right) synapses. Arrowheads indicate example puncta used to define ROIs. Data points represent the fluorescent intensity of ELKS1α within an ROI plotted against the ELKS2α signal in the same ROI. Within a single channel, individual puncta are normalized to the average intensity across all puncta (excitatory synapses: 329 ROIs/30 images/3 independent cultures; inhibitory synapses: 250/30/3). ρ: Spearman rank correlation between ELKS1α and ELKS2α.

*Figure 1 continued on next page*

*Figure 1 continued*

The following figure supplements are available for figure 1:

**Figure supplement 1.** ELKS antibody specificity.

**Figure supplement 2.** Frequency distributions of ELKS1α and ELKS2α at excitatory and inhibitory synapses.

prominently expressed at excitatory hippocampal synapses. In contrast to its roles in enhancing $Ca^{2+}$ influx and release probability at inhibitory synapses, ELKS1α/2α control the size of the RRP at excitatory hippocampal synapses. These data establish that removal of ELKS has differential, synapse-specific effects on RRP size and P: boosting RRP at excitatory synapses but enhancing P at inhibitory synapses. Using structure-function rescue experiments, we then determine the sequences within ELKS required for RRP enhancement at excitatory synapses. Surprisingly, RIM-binding sequences are dispensable for this function, but $CC_A$-$CC_C$, which include binding sites for Liprin-α and Bassoon, control excitatory RRP size. Together, these data show that ELKS selectively controls RRP at excitatory synapses through its N-terminal protein interaction motifs.

## Results

### ELKS1α and ELKS2α are enriched at excitatory hippocampal synapses

The distribution of individual ELKS proteins at excitatory and inhibitory synapses is not known. We generated ELKS2α specific antisera by immunization of rabbits (*Figure 1—figure supplement 1A–C*) and used this, in addition to an available ELKS1α specific antibody, to determine if either or both ELKS proteins are present at excitatory and inhibitory synapses. We employed immunostainings for ELKS1α and ELKS2α in cultured hippocampal neurons and analyzed their distribution using confocal microscopy. Both ELKS proteins were present at excitatory and inhibitory synapses and we observed higher intensity staining at excitatory synapses compared to inhibitory synapses for ELKS1α and ELKS2α (*Figure 1B*). A recent proteomic study of release site composition found that overall differences between excitatory and inhibitory release sites are small (*Boyken et al., 2013*). Interestingly, however, ELKS1 and ELKS2 were modestly enriched at docking sites for glutamatergic vesicles, consistent with our observation. We next examined the distribution of ELKS relative to one another to determine whether levels of individual ELKS proteins correlate positively or negatively at excitatory or inhibitory synapses. Synaptic markers for either excitatory or inhibitory synapses were used to define regions of interest (ROI) and co-stained for both ELKS1α and ELKS2α (*Figure 1C*). We found a strong positive correlation between ELKS1α and ELKS2α at excitatory and inhibitory synapses and both ELKS proteins showed a single peak in the distribution of fluorescence intensity at each synapse (*Figure 1—figure supplement 2*). Together, these data reveal that ELKS1α and ELKS2α are enriched at excitatory synapses and they suggest that synapses rich in ELKS1α are rich in ELKS2α and vice versa.

### ELKS1α and ELKS2α control neurotransmitter release at excitatory hippocampal synapses

Excitatory transmission was not affected in single ELKS2α mutants (*Kaeser et al., 2009*) and excitatory synaptic transmission was not studied in ELKS1α/ELKS2α double mutants. We thus decided to determine the function of ELKS1α/2α at excitatory synapses in cultured hippocampal neurons employing mice in which the first coding exon of each gene, *Erc1* and *Erc2,* is flanked by loxP sites (*Liu et al., 2014*; *Kaeser et al., 2009*). At 3–5 days in vitro (DIV), we infected the neurons with lentiviruses that express a cre-recombinase tagged with EGFP driven by a neuron-specific synapsin promoter (*Liu et al., 2014*) to generate ELKS1α/ELKS2α knockout neurons (cDKO). In all experiments, control neurons were genetically identical with the exception that they were infected with lentiviruses that expressed an inactive, truncated cre protein. We only analyzed cultures in which no non-infected neurons could be detected.

Using focal stimulation, we recorded action-potential evoked excitatory postsynaptic currents (EPSCs) and directly compared effects to inhibitory PSCs (IPSCs). Similar to the effect on IPSCs

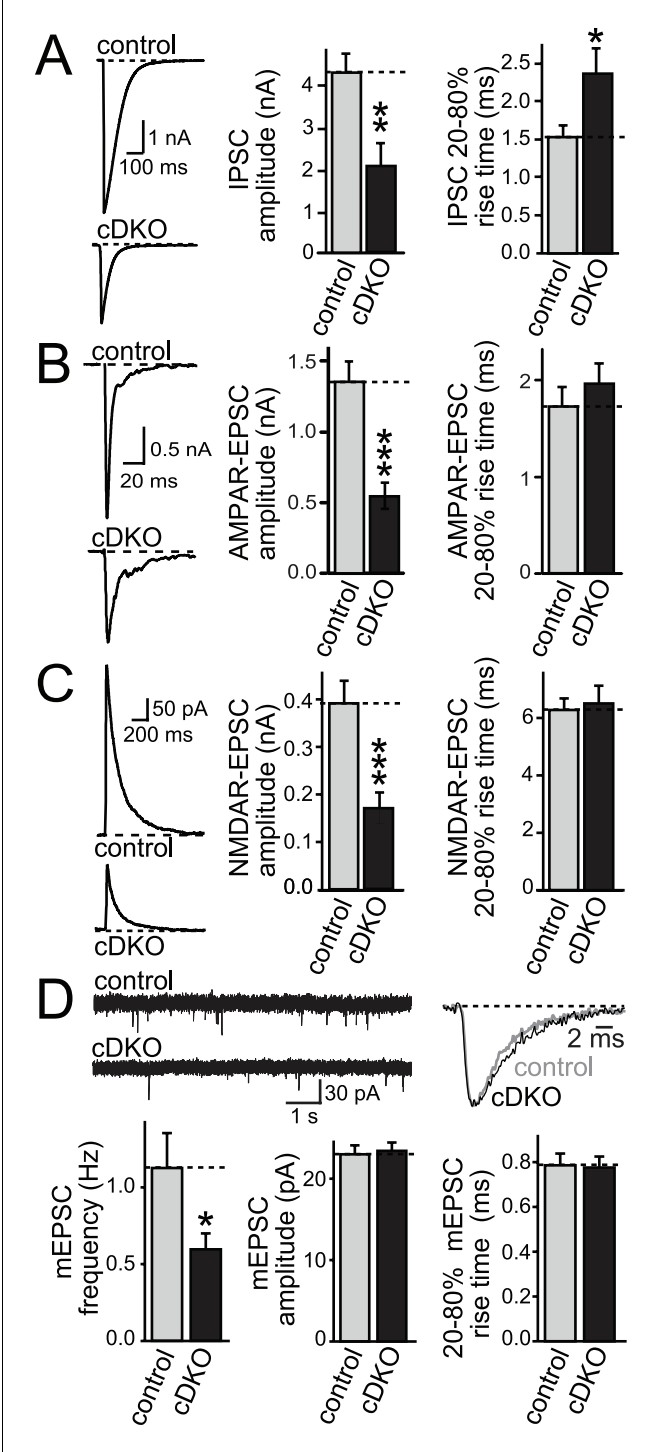

**Figure 2.** ELKS1α/2α control neurotransmitter release at excitatory synapses. (A–C) Sample traces and quantification of IPSC (A), AMPAR-EPSC (B), and NMDAR-EPSC (C) amplitudes and rise times in control and ELKS1α/2α cDKO neurons. Bar graphs show quantification of the peak amplitude (middle) and quantification of the rise time from 20% to 80% of the peak amplitude (right, A: control n = 18 cells/4 independent cultures, cDKO n = 18/4; B: control n = 18/4, cDKO n = 16/4; C: control n = 15/3, cDKO n = 15/3). (D) Sample traces (top) and quantification (bottom) of mEPSC frequency, amplitude, and 20–80% rise time (control n = 16/3, cDKO n = 15/3). Sample traces on the top left show 10 s of recording time. Sample traces on the top right are the overlayed averaged events from an individual cell in each condition normalized for amplitude. All data are means ± SEM; *p≤0.05, **p≤0.01, ***p≤0.001 as determined by Student's t test.

*Figure 2 continued on next page*

*Figure 2 continued*

The following figure supplement is available for figure 2:

**Figure supplement 1.** No change in synapse number in ELKS1α/2α cDKO cultures.

(*Figure 2A*), EPSCs were reduced to ~50% (*Figure 2B,C*). At inhibitory synapses, removal of ELKS resulted in a 30% decrease in presynaptic $Ca^{2+}$ influx which led to a reduction in P (*Liu et al., 2014*) and prolonged IPSC rise times (*Figure 2A*). When we examined the EPSC rise times, there was no effect of ELKS1α/2α cDKO (*Figure 2B,C*), suggesting that ELKS may operate differently at excitatory synapses. To determine whether the deficit in excitatory transmission was presynaptic, we recorded miniature EPSCs (mEPSCs) in the presence of TTX. mEPSC frequency was reduced by ~50%, but there was no change in mEPSC amplitude, rise time (*Figure 2D*), or decay kinetics (control τ = 5.727 ms, n = 16/3; cDKO τ = 6.321 ms, n = 15/3; p>0.05). The number and size of excitatory synapses was also unchanged (*Figure 2—figure supplement 1*). Thus, ELKS1α/2α cDKO reduces release from excitatory presynaptic nerve terminals, but the mechanisms may be different from inhibitory synapses.

## ELKS enhances the size of the RRP but not $Ca^{2+}$ influx at excitatory synapses

At inhibitory hippocampal synapses, ELKS boosts presynaptic $Ca^{2+}$ influx to increase P and to accelerate the IPSC rise. To directly test the prediction that ELKS functions differently at excitatory synapses, we employed imaging to measure the presynaptic $Ca^{2+}$ transient in response to a single action potential (*Figure 3A*). Neurons were filled with the $Ca^{2+}$ indicator Fluo-5F and a fixable Alexa-594 dye and $Ca^{2+}$ influx into presynaptic boutons was imaged during a single action potential elicited by a brief current injection through the patch pipette. This method has revealed impaired $Ca^{2+}$ influx at inhibitory ELKS deficient synapses (*Liu et al., 2014*). After the experiment, cells were fixed and stained with GAD67 antibodies to exclude GAD67 positive inhibitory neurons from the analysis. This method reliably distinguished excitatory and inhibitory neurons in hippocampal cultures (*Figure 3—figure supplement 1*). We found that there was no change in peak action potential evoked $Ca^{2+}$ influx into boutons of excitatory neurons (*Figure 3A*). At inhibitory synapses, decreased $Ca^{2+}$ influx in ELKS deficient neurons resulted in a reduction in P. Thus, our data suggest that initial P may not be affected at excitatory ELKS1α/2α cDKO synapses. Initial P is inversely correlated with paired-pulse ratios (PPR). We measured PPRs of EPSC amplitudes by monitoring *N*-Methyl-*D*-aspartate receptor EPSCs (NMDAR-EPSCs) to circumvent the strong reverberant activity that is present in cultured networks when α-amino-3-hydroxy-5-methyl-4-isoxazolepropionic acid (AMPA) receptors are not blocked (*Maximov et al., 2007*). Consistent with the $Ca^{2+}$ imaging data, but different from inhibitory synapses (*Liu et al., 2014*), there were no changes in PPR in ELKS cDKO neurons across all tested interstimulus intervals (*Figure 3B*). These experiments establish that defects in $Ca^{2+}$ influx and P are unlikely to explain impaired neurotransmitter release at excitatory synapses of ELKS cDKO neurons.

We next tested whether the size of the RRP was changed at excitatory synapses. We stimulated release of the entire RRP using a hypertonic sucrose solution (500 mOsm) and quantified RRP size by integrating the total charge transfer during the first ten seconds of the response (*Rosenmund and Stevens, 1996*). Although the hypertonic stimulus is non-physiological, this method has been used as a snapshot measurement of RRP in cultured neurons and has been insightful for genotype comparisons and for dissecting molecular mechanisms of RRP control (*Augustin et al., 1999*; *Deng et al., 2011*; *Neher, 2015*; *Rosenmund and Stevens, 1996*). At excitatory synapses, the RRP was reduced by ~40% (*Figure 4A*) in ELKS cDKO neurons, providing an explanation for a reduction in release in the absence of changes in P at excitatory ELKS1α/2α cDKO synapses. Short action potential trains (10 stimuli at 20 Hz), a more physiological stimulus, resulted in a 50% reduction in charge transfer during the stimulus train in ELKS1α/2α cDKO neurons compared to control neurons (*Figure 4B*). The delayed charge transfer after stimulation ended was similarly reduced. These data are consistent with a reduced number of RRP vesicles available for release in response to a brief action potential train.

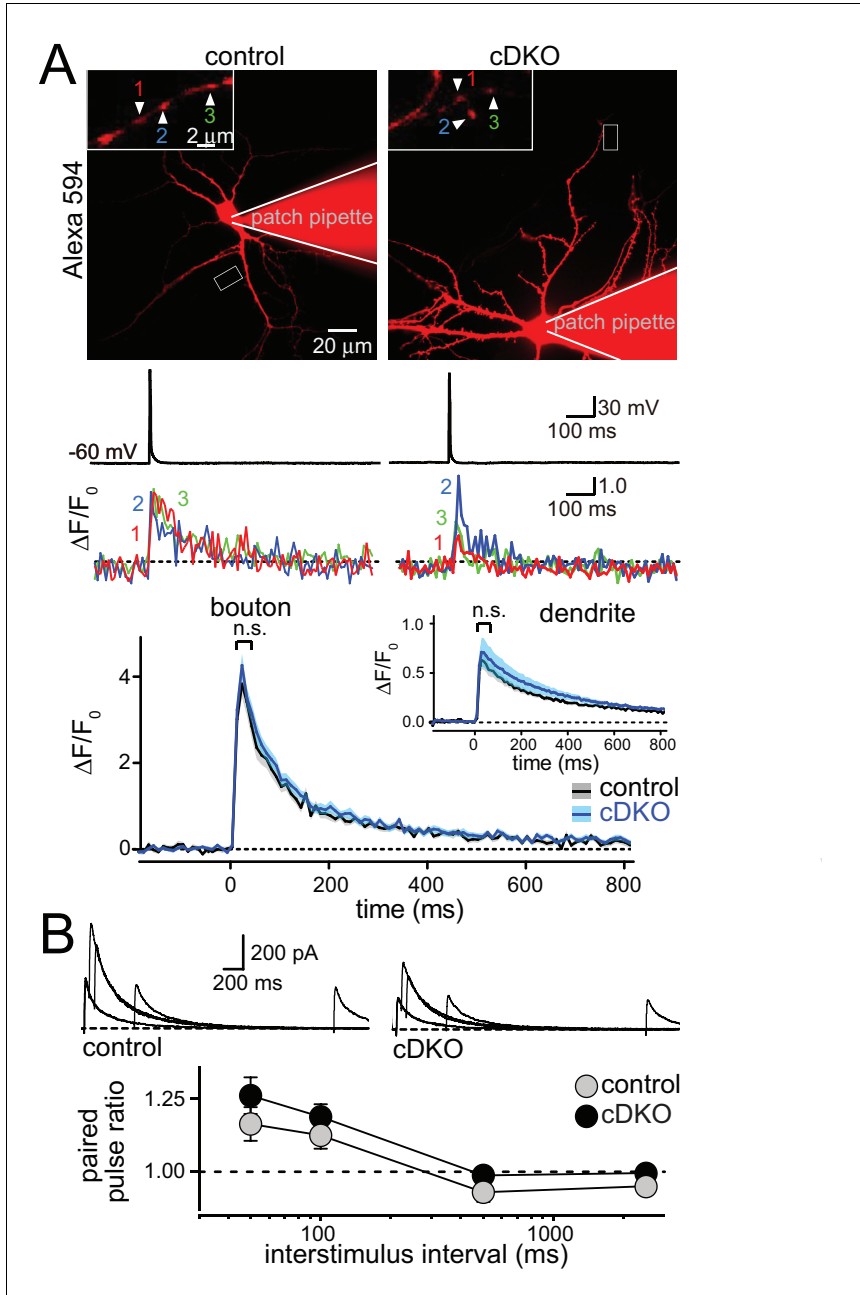

**Figure 3.** ELKS1α/2α do not control Ca²⁺ influx and release probability in excitatory nerve terminals. (**A**) Sample images (top, imaged boutons are numbered and color coded), action potential and imaging traces (middle) and summary plots (bottom) of single action potential-induced Ca²⁺ transients imaged by Fluo-5F fluorescence in presynaptic boutons are shown, the inset shows the same plot for dendrites. Data are shown as mean (line) ± SEM (shaded area) and analyzed by two-way ANOVA: genotype, n.s.; time, ***p<0.001; interaction, n.s. (Boutons: control n = 60 boutons/6 cells /4 independent cultures, cDKO n = 80/8/4; dendrites: control n = 6 dendrites/6 cells/4 independent cultures, cDKO n = 8/8/4). (**B**) Sample traces (top) and quantification of paired pulse ratios (PPRs, bottom) of evoked NMDAR-EPSCs. Example traces showing overlayed responses to pairs of stimuli at 50, 100, 500, and 2500 ms interstimulus intervals. PPRs (amplitude 2/amplitude 1) are plotted against the interstimulus interval. Significance as analyzed by two-way ANOVA: genotype, n.s.; interstimulus interval, ***p<0.001; interaction, n.s. (control n = 17 cells/3 independent cultures, cDKO n = 19/3).

The following figure supplement is available for figure 3:

**Figure supplement 1.** Post-hoc identification of excitatory neurons after presynaptic Ca²⁺ imaging.

One possible explanation for the phenotypic differences between inhibitory transmission as observed earlier (*Liu et al., 2014*) and excitatory transmission as described here is that removal of ELKS has effects that vary between cultures and over time, causing some cultures to have more pronounced effects on RRP size and others to exhibit changes in P. To control for this possibility, we conducted a series of experiments analyzing RRP size and P (as measured by PPRs) for inhibitory and excitatory transmission in the same cultures and matching sample size for all conditions. Both inhibitory and excitatory synapses had a reduction in action potential-evoked EPSC and IPSC amplitudes (*Figure 5—figure supplement 1*). At inhibitory synapses, ELKS1α/2α cDKO increased the PPR at low interstimulus intervals but had no significant effect on RRP size, consistent with a reduction of P (*Figure 5A,C*) and with our previous study (*Liu et al., 2014*). In contrast, excitatory synapses in the same cultures showed no change in PPR but a reduction in RRP size (*Figure 5B,D*). Notably, there was a small, non-significant trend towards a reduced RRP at inhibitory synapses in this experiment. To further rule out that there is a reproducible RRP reduction at inhibitory hippocampal synapses, we conducted a second experiment measuring RRP with a slightly different protocol as described in the methods section and used in (*Liu et al., 2014*). Again, no change in RRP was detected (*Figure 5—figure supplement 2*). Thus, in three independent measurements (*Figure 5C*, *Figure 5—figure supplement 2* and Figure 7F in *Liu et al., 2014*), no reduction in the inhibitory RRP could be detected. These data confirm our previous experiments and establish synapse-specific roles for ELKS in neurotransmitter release: at excitatory synapses ELKS1α/2α primarily boost the RRP (*Figures 1–5*), whereas at inhibitory synapses ELKS1α/2α mainly control action potential induced $Ca^{2+}$ influx to enhance P (also see Figures 5 and 8 in *Liu et al., 2014*).

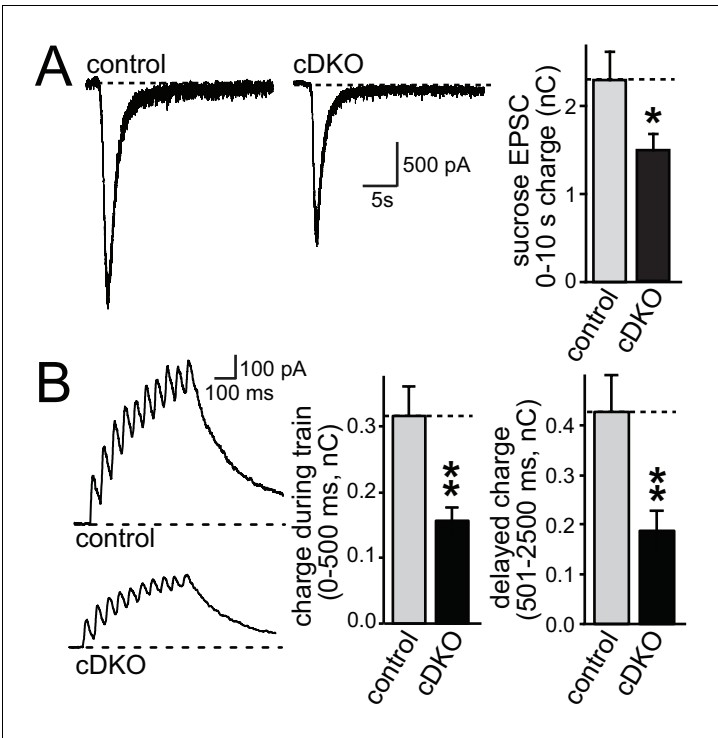

**Figure 4.** ELKS1α/2α control RRP at excitatory synapses. (**A**) Sample traces showing AMPAR-EPSCs in response to superfusion with 500 mOsm sucrose (left). The bar graph on the right shows the AMPAR-EPSC charge transfer, quantified as the area under the curve during the first ten seconds of the response (control n = 30/3, cDKO n = 29/3). (**B**) Sample traces (left) of NMDAR-EPSCs during a short action potential train (10 stimuli at 20 Hz). The charge transfer during the train and in the two seconds immediately after the train is quantified separately on the right (control n = 14/3, cDKO n = 14/3). All data are means ± SEM; *p≤0.05, **p≤0.01 as determined by Student's t test.

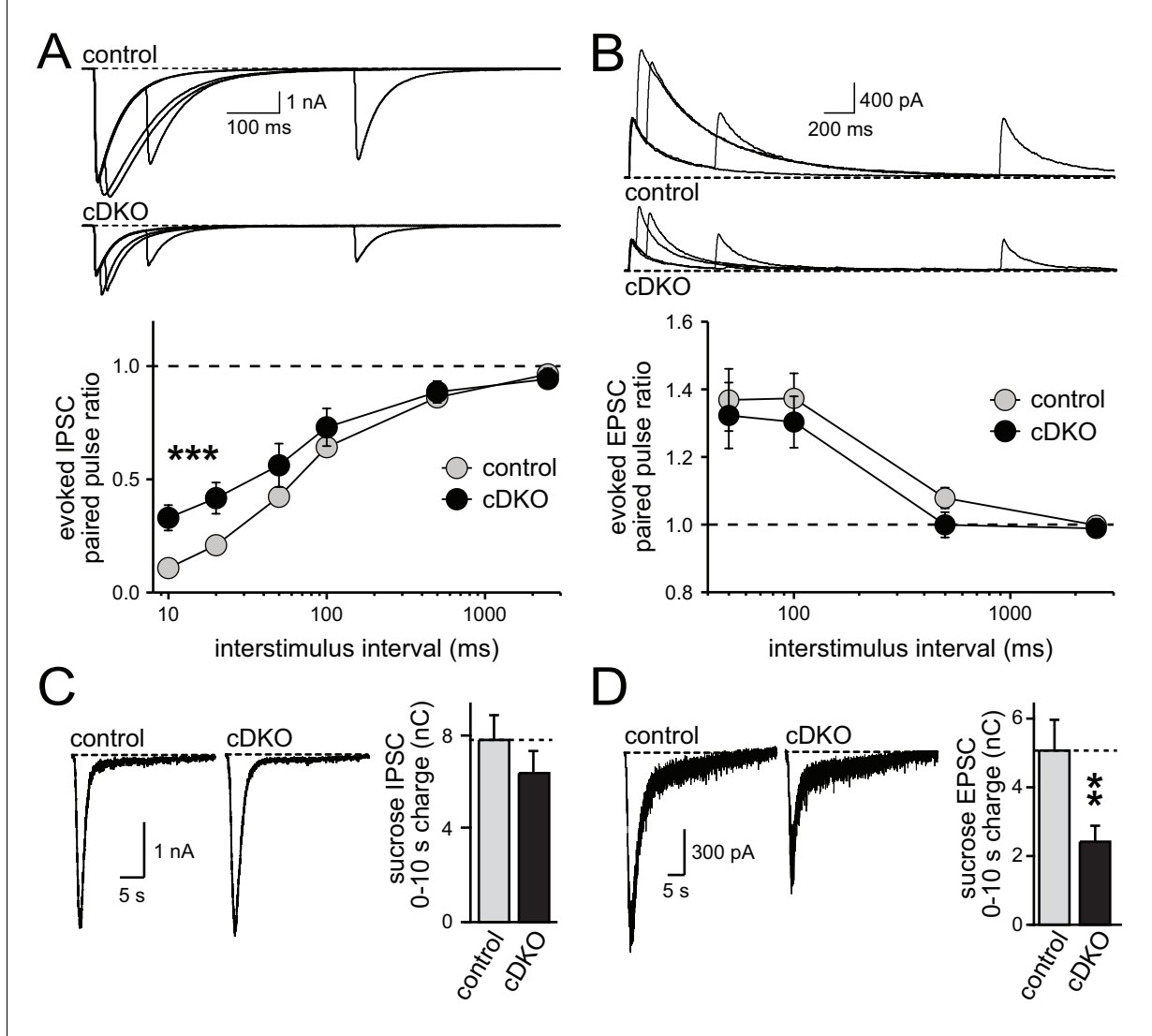

**Figure 5.** Direct comparison of IPSC and EPSC phenotypes of ELKS1α/2α cDKO. (**A**) Example traces (top) showing overlayed IPSC responses to pairs of stimuli at 10, 20, 100, and 500 ms interstimulus intervals. Paired-pulse ratios (amplitude 2/amplitude 1) are plotted against the interstimulus interval (10, 20, 50, 100, 500, and 2500 ms intervals). Significance as analyzed by two-way ANOVA: genotype, ***p<0.001; interstimulus interval, ***p<0.001; interaction, n.s. Holm-Sidak post-hoc test: 10 ms, *p<0.05; 20 ms, *p<0.05 (control n = 15 cells/3 independent cultures, cDKO n = 15/3). (**B**) Example traces (top) showing overlayed EPSC responses to pairs of stimuli at 50, 100, 500, and 2500 ms interstimulus intervals. Paired-pulse ratios (amplitude 2/ amplitude 1) are plotted against the interstimulus interval (50, 100, 500, and 2500 ms intervals). Significance as analyzed by two-way ANOVA: genotype, n.s.; interstimulus interval, ***p<0.001; interaction, n.s. (control n = 15/3, cDKO n = 15/3). (**C**) Sample traces showing IPSCs in response to superfusion with 500 mOsm sucrose (left) and quantification (right) of IPSC charge transfer during the first ten seconds of the response (control n = 15/3, cDKO n = 15/3). (**D**) Sample traces showing EPSCs in response to superfusion with 500 mOsm sucrose (left) and quantification (right) of EPSC charge transfer during the first ten seconds of the response (control n = 15/3, cDKO n = 15/3). Data are means ± SEM; **p≤0.01 as determined by Student's t test.

The following figure supplements are available for figure 5:

**Figure supplement 1.** Direct comparison of IPSC and EPSC amplitudes of ELKS1α/2α cDKO.

**Figure supplement 2.** Inhibitory RRP size in ELKS1α/2α cDKO neurons.

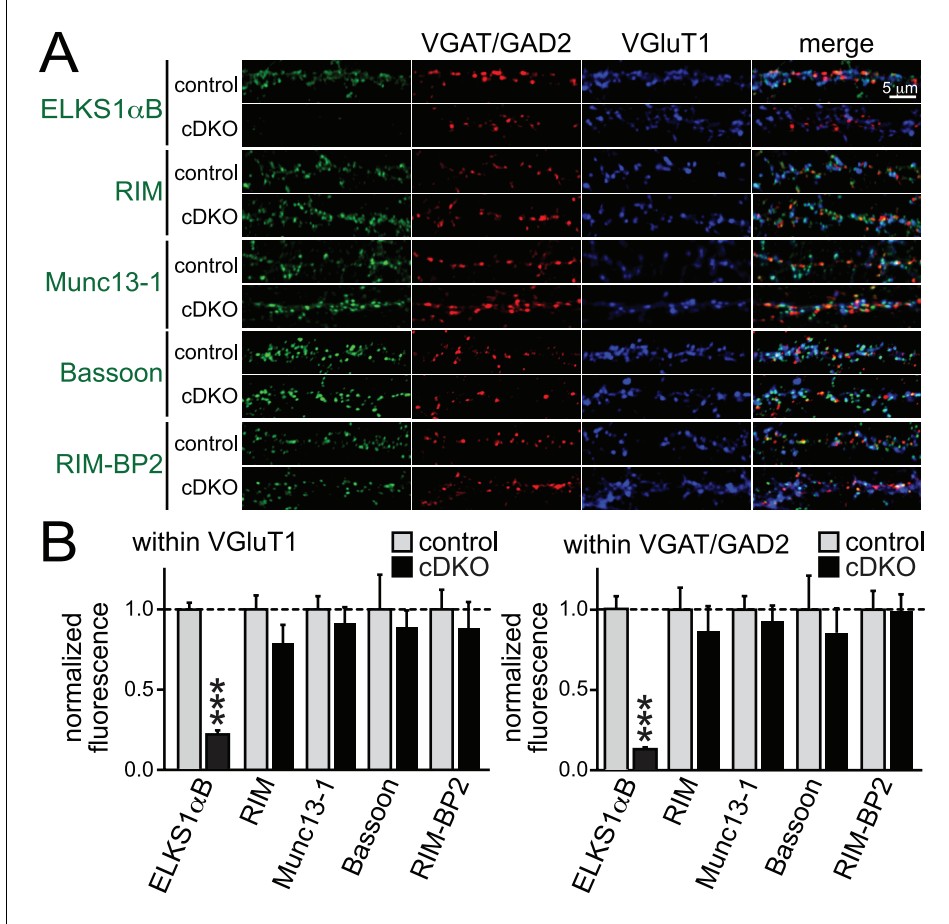

**Figure 6.** Active zone composition in ELKS1α/2α cDKO synapses. (**A**) Sample images of control and ELKS1α/2α cDKO neurons stained with antibodies against active zone proteins. Inhibitory synapses were marked with VGAT (for ELKS, RIM and Bassoon) or GAD2 (for Munc13-1 and RIM-BP2), excitatory synapses were marked with VGluT1. (**B**) Quantification of active zone proteins within ROIs defined by excitatory (left) or inhibitory (right) synaptic markers (RIM: control n = 4 independent cultures, cDKO n = 4; Munc13-1: control n = 4, cDKO n = 4; Bassoon: control n = 3, cDKO n = 3; RIM-BP2: control n = 3, cDKO n = 3; in each culture, 10 images were averaged per culture and genotype). Data are means ± SEM; ***p≤0.001 as determined by Student's t test.

## Removal of ELKS1α/2α does not lead to loss of presynaptic priming proteins

Given the hypothesis that ELKS is a presynaptic scaffold, we set out to test the possibility that loss of ELKS1α/2α changes presynaptic levels of proteins involved in controlling RRP size at excitatory synapses. RIM and Munc13-1 were of particular interest, since both proteins localize to the active zone and knockouts of either protein have strong RRP impairments at excitatory hippocampal synapses (*Augustin et al., 1999*; *Calakos et al., 2004*; *Kaeser et al., 2011*). Using confocal microscopy, we quantified the synaptic levels of the active zone proteins ELKS, RIM, Munc13-1, Bassoon, and RIM-BP2 in cDKO and control neurons. With the exception of ELKS, no protein was significantly reduced at excitatory or inhibitory synapses (*Figure 6*). These data are consistent with normal electron microscopic appearance of ELKS1α/2α cDKO synapses (*Liu et al., 2014*) and indicate that ELKS is not essential to recruit the priming proteins RIM and Munc13-1 to the presynaptic nerve terminal.

## ELKS controls RRP through the N-terminal coiled-coil domains

Since ELKS1α/2α cDKO caused no detectable structural changes at the active zone, we instead turned to electrophysiological rescue experiments to determine how ELKS might control RRP size. ELKS binds directly to the PDZ domain of RIM through its four C-terminal residues and may have

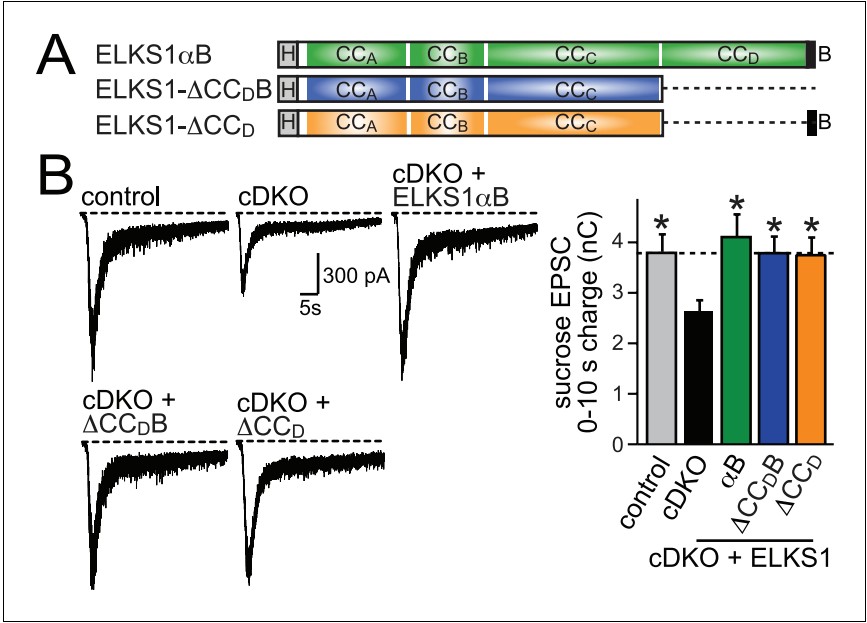

**Figure 7.** C-terminal ELKS sequences do not support RRP in ELKS1α/2α cDKO neurons. (**A**) Schematic of ELKS1 rescue constructs; CC$_{A-D}$: coiled-coil regions A-D, B: PDZ-binding motif; H: human influenza hemagglutinin (HA) tag, deleted sequences are illustrated as dashed lines. (**B**) Sample traces (left) and quantification (right) of the AMPAR-EPSC charge in response to hypertonic sucrose application, measured as area under the curve during the first ten seconds after the start of the stimulus (control n = 26 cells/5 independent cultures, cDKO n = 27/5, cDKO + ELKS1αB n = 21/5, cDKO + ELKS1-△CC$_D$B n = 23/5, cDKO + ELKS1-△CC$_D$ n = 21/5). All data are means ± SEM; *p≤0.05 as determined by one-way ANOVA followed by Holm-Sidak multiple comparisons post-hoc test comparing each condition to cDKO.

The following figure supplement is available for figure 7:

**Figure supplement 1.** Expression and localization of ELKS1 C-terminal rescue constructs.

roles in active zone anchoring of RIM (*Lu et al., 2005*; *Wang et al., 2002*; *Ohtsuka et al., 2002*; *Kaeser et al., 2009*). Peptide injection experiments using the RIM-binding domain of ELKS support a role of the RIM-ELKS interactions in release (*Takao-Rikitsu et al., 2004*). The CC$_D$ domain, which is adjacent to the RIM binding sequence, also binds to Ca$^{2+}$ channel β4 subunits (*Kiyonaka et al., 2012*). These interactions suggest that the C-terminus of ELKS may tether ELKS to the active zone to support its functions in release.

We designed lentiviral rescue constructs in which we expressed either full length ELKS1αB or mutant ELKS1 that lacks either the entire C-terminal region including the PDZ binding motif (△CC$_D$B) or just CC$_D$ (△CC$_D$) (*Figure 7A*). All three rescue constructs expressed at levels similar to wild type ELKS1α and localized to synapses (*Figure 7—figure supplement 1*), suggesting that the ELKS C-terminal regions are not necessary for ELKS localization. Surprisingly, ELKS1αB, ELKS1-△CC$_D$B, and ELKS1-△CC$_D$ were sufficient to rescue excitatory RRP size in ELKS cDKO neurons (*Figure 7B*). Thus, the ELKS C-terminal domains that bind to RIM and other presynaptic proteins are not necessary for its control of the RRP at excitatory synapses. Altogether, the C-terminal ELKS domains are unlikely to act as a central scaffolding hub at the active zone.

We next decided to take an unbiased approach and systematically tested all ELKS protein interaction sites covering the entire sequence of ELKS1αB in rescue experiments. We generated rescue constructs lacking N-terminal coiled-coils (corresponding to ELKS1βB) or the central coiled-coil region (△CC$_C$). ELKS1αB and ELKS1-△CC$_D$B were used as positive rescue controls (*Figure 8A*). All rescue constructs were successfully expressed, albeit at variable levels (*Figure 8—figure supplement 1A*). Compellingly, neither ELKS1βB nor ELKS1-△CC$_C$ rescued the RRP in ELKS1α/2α cDKO neurons (*Figure 8B*). The lack of rescue could be due to either a local function of the deleted coiled-

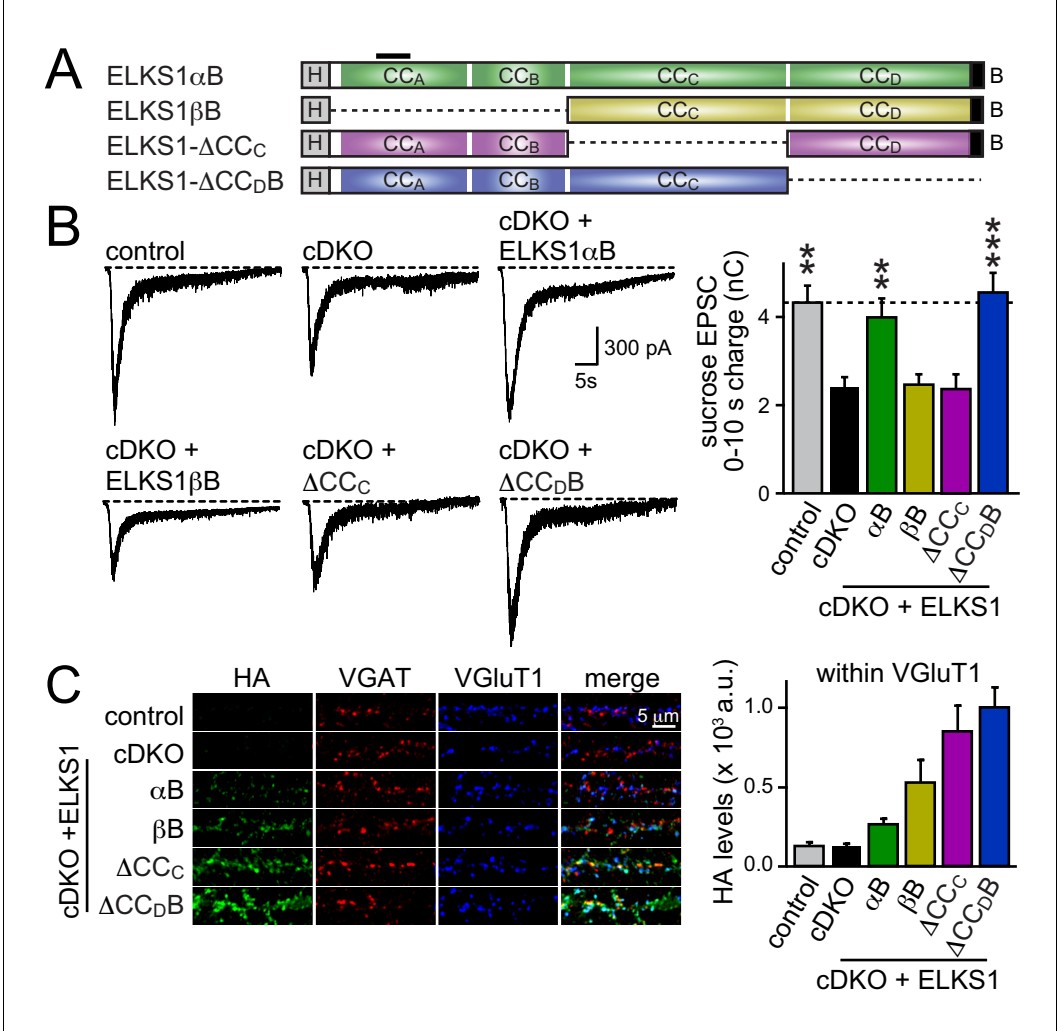

**Figure 8.** N-terminal coiled-coil domains of ELKS control RRP size at excitatory synapses. (**A**) Schematic of ELKS1 rescue constructs; CC$_{A-D}$: coiled-coil regions A-D, B: PDZ-binding motif; H: human influenza hemagglutinin (HA) tag, black bar: antigen recognized by the ELKS1α antibody (E-1) used in figure supplement 1C. Deleted sequences are illustrated as dashed lines, (**B**) sample traces (left) and quantification (right) of the AMPAR-EPSC charge in response to hypertonic sucrose application, measured as area under the curve during the first ten seconds after the start of the stimulus (control n = 21 cells/4 independent cultures, cDKO n = 22/4, cDKO + ELKS1αB n = 19/4, cDKO + ELKS1βB n = 18/4, cDKO + ELKS1-△CC$_C$ n = 20/4, cDKO + ELKS1-△CC$_D$B n = 21/4). (**C**) Sample images (left) of control, cDKO, and cDKO + rescue neurons stained with antibodies against HA. Quantification (right) of HA fluorescent intensity within ROIs defined by VGluT1 (control n = 3 independent cultures, cDKO n = 3, cDKO + ELKS1αB n = 3, cDKO + ELKS1βB n = 3, cDKO + ELKS1-△CC$_C$ n = 3, cDKO + ELKS1-△CC$_D$B n = 3, 5–10 images were averaged per culture and genotype). All data are means ± SEM; **p≤0.01, ***p≤0.001 as determined by one-way ANOVA followed by Holm-Sidak multiple comparisons post-hoc test comparing each condition to cDKO.

The following figure supplements are available for figure 8:

**Figure supplement 1.** Expression and localization of ELKS1 full length rescue constructs.

**Figure supplement 2.** Rescue of action potential evoked IPSCs with ELKS1βB.

coil regions or to a role for these regions in localizing ELKS to synapses. We distinguished between these possibilities by assessing the localization of each rescue construct using confocal microscopy. Since ELKS1βB lacks the antigen recognized by our ELKS1 antibody we stained for either the HA tag

included in all rescue proteins (*Figure 8C* and *Figure 8—figure supplement 1B*) or for ELKS1α (*Figure 8—figure supplement 1C–E*). Synaptic expression of the rescue constructs ranged from 50–200% of wild-type levels of ELKS1. Importantly, all constructs localized to synapses. Thus vertebrate ELKS1 can be localized to synapses likely through multiple redundant interactions. While synaptic levels of rescue ELKS1αB were low, it rescued entirely. In contrast, ELKS1βB and ELKS1-△CC$_C$ both failed to rescue, but were expressed above ELKS1αB levels. Removal of ELKS reveals differential impairments of RRP and P at excitatory and inhibitory synapses, respectively (*Figure 5*). To test whether such differential effects are also reflected in the ELKS sequences that mediate rescue at each synapse, we tested whether ELKS1βB, an ELKS isoform that failed to rescue the excitatory RRP, was sufficient to restore inhibitory synaptic transmission (*Figure 8—figure supplement 2*). ELKS1βB was able to rescue action-potential evoked IPSC amplitudes, establishing that ELKS' role in controlling P at inhibitory synapses does not require the N-terminal coiled-coil regions that control the excitatory RRP. Together, these experiments establish that the N-terminal coiled-coil sequences of ELKS1α are necessary for ELKS' role in enhancing the RRP at excitatory synapses, but not necessary for localizing ELKS to synapses.

## Discussion

Our findings define a new functional role for ELKS in setting RRP size at excitatory hippocampal synapses. They extend beyond previous understanding of the role of ELKS in supporting presynaptic Ca$^{2+}$ influx at inhibitory synapses (*Liu et al., 2014*) and demonstrate that this is not the primary function across all synapses. Most current molecular models of active zone function imply that active zones operate essentially identically across synapses despite the notion that the parameters they control, RRP and P, vary greatly between different types of synapses. Thus far, this assumption has proven principally true for genetic analyses of active zone protein function. For example, it is widely accepted that Munc13 is required at all synapses to prime vesicles and RIMs anchor and activate Munc13 to support priming (*Varoqueaux et al., 2002*; *Andrews-Zwilling et al., 2006*; *Deng et al., 2011*; *Calakos et al., 2004*; *Augustin et al., 1999*). Our findings are a starting point for the dissection of synapse-specific architecture and function of the active zone.

One possibility to explain synapse-specific roles is that ELKS1α and ELKS2α account for different functions, and that they are localized to specific subsets of synapses. Such differences have been observed for roles of Munc13-1 and bMunc13-2 in short-term plasticity (*Rosenmund et al., 2002*). Our data make this possibility unlikely in the case of ELKS, because there is a strong positive correlation between ELKS1α and ELKS2α levels at all synapses. Furthermore, no ELKS isoform-specific protein interactions are described in the literature, and the interaction sequences are generally well conserved between ELKS1α and ELKS2α (*Monier et al., 2002*; *Wang et al., 2002*).

Another possibility is that interaction partners of ELKS that mediate RRP control are distributed in a synapse specific fashion, and ELKS protein interactions at the active zone may be engaged differentially depending on the presence of specific interacting proteins. This model is supported by the observation that the sequence requirements for rescue of the excitatory RRP or the IPSC are different. ELKS interacts in vitro with many active zone proteins (*Ohtsuka et al., 2002*; *Wang et al., 2002*; *Schoch and Gundelfinger, 2006*). We find that ELKS N-terminal, but not C-terminal, sequences are required for RRP control. These N-terminal sequences have been shown to bind to Liprin-α via CC$_A$-CC$_C$ (*Ko et al., 2003*), to Bassoon via CC$_C$ (*Takao-Rikitsu et al., 2004*), and they may mediate binding to Rab6 (*Monier et al., 2002*). Because constructs including CC$_C$ but lacking CC$_{A/B}$ localize to synapses but do not to rescue, one possible explanation is that Liprin-α binding is required for ELKS function in controlling the excitatory RRP. Vertebrate genomes contain four genes encoding Liprin-α, but it is not known whether Liprin-α isoforms are differentially distributed to specific active zones. In fact, it is currently not clear for any of the vertebrate Liprin-α isoforms whether there is a tight association with the presynaptic active zone (*Zurner et al., 2011*; *Spangler et al., 2011*). Functional roles of vertebrate Liprin-α in synaptic transmission are not well understood, but a recent study supports presynaptic roles for Liprin-α2 at hippocampal synapses (*Spangler et al., 2013*). In invertebrates, Liprin-α/syd-2 has synaptogenic activities, and effects of a gain of function mutation in syd-2 require ELKS (*Dai et al., 2006*; *Patel et al., 2006*), indicating important synaptic roles for Liprin-α/ELKS interactions. Thus, these previous studies are consistent with the hypothesis that ELKS controls RRP through Liprin-α. Nevertheless, it is possible that binding to Bassoon, Rab6,

or unknown proteins mediate the role of ELKS in enhancing RRP. A recent proteomic analysis of vesicle docking complexes revealed only modest differences in the composition of docking sites at glutamatergic compared to GABAergic synapses (*Boyken et al., 2013*). Interestingly, however, Bassoon and ELKS were found to be enriched in glutamatergic synapses in this study, perhaps supporting a role for these proteins for promoting excitatory transmission in concert with one another. Furthermore, a recent study showed that Bassoon regulates release indirectly via RIM-BP through a specific $Ca^{2+}$ channel subunit, which could result in a synapse-specific roles for Bassoon (*Davydova et al., 2014*). In summary, these studies provide support for the hypothesis that ELKS may operate with Bassoon or Liprin-α to promote release in a synapse-specific fashion.

Our studies also reveal that ELKS1α and ELKS2α are not required for active zone assembly and maintenance at hippocampal synapses, consistent with studies in *C. elegans* (*Deken et al., 2005*; *Patel et al., 2006*). At the *D. melanogaster* neuromuscular junction, however, Brp is essential for the formation of T-bars. The ELKS homology of Brp is limited to the N-terminal half of the protein (*Kittel et al., 2006*; *Monier et al., 2002*), making it possible that such strong scaffolding functions are specific to Brp and absent in ELKS. Alternatively, an ELKS scaffolding function may not be detected in our experiments due to redundant scaffolding activities in other proteins, for example β-ELKS or RIM (*Schoch et al., 2002*; *Kaeser et al., 2009*; *Liu et al., 2014*).

Several studies suggested that there is a good correlation between the number of docked vesicles and the size of the RRP at hippocampal synapses (*Schikorski and Stevens, 2001*; *Imig et al., 2014*), but electron microscopic analysis of ELKS1α/2α cDKO synapses using conventional fixation has not revealed a deficit in vesicle docking (*Liu et al., 2014*). Recent work has achieved better resolution in the analysis of docking by combining high-pressure freezing with EM tomography (*Imig et al., 2014*; *Siksou et al., 2007*). We can currently not rule out the possibility that ELKS1α/2α cDKO causes a subtle vesicle docking phenotype undetectable by the methods used in our previous work. An alternative hypothesis to a reduction of RRP at each excitatory synapse is the possibility that a subpopulation of excitatory synapses is very strongly affected by the removal of ELKS. Because the distribution of ELKS fluorescence intensity across excitatory synapses has a single peak, it is unlikely that ELKS only operates at a subset of excitatory synapses and that these synapses are silent in the absence of ELKS. However, additional work at clearly defined synapse populations with less heterogeneity than mixed hippocampal cultures will be necessary to address these questions.

Finally, although ELKS1α/2α cDKO neurons do not reveal an RRP deficit as measured by hypertonic stimuli at inhibitory synapses (*Figure 5* and *Liu et al., 2014*), ELKS is known to regulate RRP at these synapses. Enhancing P by increasing extracellular $Ca^{2+}$ cannot completely rescue synaptic transmission at inhibitory ELKS1α/2α cDKO synapses (*Liu et al., 2014*), and removal of ELKS2α alone leads to an increase in the RRP at inhibitory synapses (*Kaeser et al., 2009*). These experiments suggest that there may be molecular regulation of the RRP at inhibitory synapses through interplay between ELKS1 and ELKS2, which are both present at inhibitory active zones.

In the long-term, it will be important to understand how the synapse-specific molecular control of RRP and P contribute to circuit function. Human genetic experiments reveal that mutations in *ERC1/ELKS1* may contribute to autism spectrum disorders (*Silva et al., 2014*), and it is possible that the pathophysiology arises from synapse-specific misregulation of neurotransmitter release.

## Materials and methods

### Mouse lines

All experiments using mice were performed according to institutional guidelines at Harvard University. Conditional double knockout (cDKO) mice that remove the ELKS1α/2α proteins were generated by crossing conditional knockout mice for the *Erc1* ([*Liu et al., 2014*] RRID:IMSR_JAX:015830) and *Erc2* ([*Kaeser et al., 2009*] RRID:IMSR_JAX:015831) genes. ELKS1α/2α cDKO mice were maintained as double homozygote line.

### Generation of antibodies

ELKS2α specific antibodies were raised in rabbits using an ELKS2 peptide ([109]LSHTDVLSYTDQ[120]). Peptides were synthesized and conjugated to keyhole lympet hemocynanin (KLH) via an N-terminal

cysteine residue. Rabbits were inoculated at Cocalico Biologicals with KLH-conjugated ELKS2 peptides and given booster injections every two weeks, and bleeds were collected every three weeks. Sera were screened against protein samples harvested from cultured neurons, brain homogenate, and transfected HEK cells expressing either ELKS1αB or ELKS2αB. β-actin was used as a loading control. The serum with the highest immunoreactivity (rabbit 1029, bleed 5) against ELKS2 was affinity purified with the ELKS2 peptide coupled to an affinity column and used at 1:100 dilution.

## Cell cultures and lentiviral infection

Primary mouse hippocampal cultures from newborn pups were generated as previously described (*Kaeser et al., 2008*, *2011*; *Maximov et al., 2007*). All lentiviruses were produced in HEK293T cells by Ca$^{2+}$ phosphate transfection. Neurons were infected with viruses that express cre recombinase or an inactive cre truncation mutant under the human synapsin promoter (*Liu et al., 2014*). Neuronal cultures were infected with 125–250 µl of HEK cell supernatant at 3–5 days in vitro (DIV). Infection efficiency was monitored by an EGFP tag attached to nuclear cre, and only cultures in which no non-infected cells were found were used for experiments. Expression of rescue proteins was achieved with a second lentivirus driven in neurons by a human synapsin promoter and applied to the neurons at DIV 3. Expression of rescue proteins was monitored by Western blotting and by immunostaining as described below.

## Immunofluorescence stainings and confocal imaging of cultured neurons

Neurons were fixed in 4% paraformaldehyde/phosphate-buffered saline, permeabilized in 0.1% Triton X-100/3% bovine serum albumin/phosphate-buffered saline, and incubated in primary antibodies overnight. The following primary antibodies were used: E-1 (1:500, RRID:AB_10841908), mouse monoclonal antibody, binds to ELKS1α isoforms (ELKS1αB, ELKS1αA); 1029 (1:100, custom rabbit polyclonal antibody generated against ELKS2α ([109]LSHTDVLSYTDQ[120]); mouse anti-RIM (1:500, RRID:AB_10611855); rabbit anti-Munc13-1 (1:5000; a gift from Dr. Nils Brose); mouse anti-Bassoon (1:500, RRID:AB_11181058); rabbit anti-RIM-BP2 (1:500, SySy, #316103); rabbit anti-VGAT (1:500, RRID:AB_887869); guinea pig anti-VGAT (1:500, RRID:AB_887873); mouse anti-GAD2 (1:500, RRID: AB_2107894, also called GAD65); guinea pig anti-VGluT1 (1:500, RRID:AB_887878); mouse anti-GAD1 (1:1000, RRID:AB_2278725, also called GAD67). Secondary antibodies conjugated to Alexa Fluor 488, 546, or 633 were used for detection. Images were acquired on Olympus FV1000 or FV1200 confocal microscopes with 60x oil immersion objectives with 1.4 numerical aperture, the pinhole was set to one airy unit, and identical settings were applied to all samples within an experiment. Single confocal sections were analyzed in ImageJ software (NIH). Background was subtracted using the rolling ball method with a radius of 2 µm. For quantification of synaptic protein levels, regions of interest (ROIs) were defined using VGAT, GAD2, or VGluT1 puncta and the average intensity of the protein of interest (in the 488 channel) inside those ROIs was quantified. In *Figure 1*, since differences in antibody affinity make raw fluorescence values between two different antibodies non-comparable, individual data points were calculated for each channel (ELKS1α or ELKS2α) by normalizing the fluorescence intensity within a single ROI to the average intensity across all ROIs. In *Figure 6*, the intensity of the ELKS1α, RIM, Bassoon, and RIM-BP2 staining in cDKO neurons was normalized to the staining in control neurons. In all other figures where only ELKS1α staining is quantified, data are expressed in arbitrary units (a.u.). When necessary, representative images were enhanced for brightness and contrast to facilitate visual inspection; all such changes were made after analysis and were made identically for all experimental conditions. All quantitative data are derived from ≥3 cultures; 5–10 fields of view were quantified per culture per genotype. For all image acquisition and analyses comparing two or more conditions, the experimenter was blind to the condition.

## Western blotting

Western blotting was performed according to standard protocols. After SDS-Page electrophoresis, gels were transferred onto nitrocellulose membranes and blocked in 10% (w/v) non-fat milk/5% (v/v) goat serum. Membranes were incubated with primary antibodies in 5% (w/v) non-fat milk/5% (v/v) goat serum for two hours at room temperature or overnight at 4°C. The following primary antibodies were used: mouse anti-ELKS1α (1:1000; E-1, RRID: AB_10841908), rabbit anti-ELKS2α (1:500; custom

antibody 1029), rabbit anti-ELKS1/2 (1:2000; P224, gift of Dr. Thomas Südhof), mouse anti-β-actin (1:2000; RRID: AB_476692). After washing, membranes were incubated with HRP-conjugated secondary antibodies in 5% (w/v) non-fat milk/5% (v/v) goat serum for one hour at room temperature, and chemiluminescence was used for detection after washing.

## Electrophysiology

Electrophysiological recordings in cultured hippocampal neurons were performed as described (*Kaeser et al., 2008*, *2009*, *2011*; *Maximov et al., 2007*; *Liu et al., 2014*) at DIV 15–19. The extracellular solution contained (in mM): 140 NaCl, 5 KCl, 2 CaCl$_2$, 2 MgCl$_2$,10 HEPES-NaOH (pH 7.4), 10 Glucose (~310 mOsm). For evoked NMDAR excitatory postsynaptic currents (EPSCs) picrotoxin (PTX, 50 µM) and 6-Cyano-7-nitroquinoxaline-2,3-dione (CNQX, 20 µM) were added to the bath. For miniature EPSC recordings and RRP measurements tetrodotoxin (TTX, 1 µM) was added to block action potentials, in addition to PTX (50 µM) for EPSCs or D-(-)-2-Amino-5-phosphonopentanoic acid (APV, 50 µM) and CNQX (20 µM) for IPSCs. All recordings were performed in whole cell patch clamp configuration at room temperature. Glass pipettes for were pulled at 2–4 MΩ and filled with intracellular solutions containing (in mM) for EPSC recordings: 120 Cs-methanesulfonate, 10 EGTA, 2 MgCl$_2$, 10 HEPES-CsOH (pH 7.4), 4 Na$_2$-ATP, 1 Na-GTP, 4 QX314-Cl (~300 mOsm) and for IPSC recordings in *Figure 1*: 120 CsCl, 5 NaCl, 10 EGTA, 1 MgCl$_2$, 10 Sucrose, 10 Hepes-CsOH (pH 7.4), 4 Mg-ATP, 0.4 GTP, 4 QX314-Cl (~300 mOsm) and *Figure 5*: 40 CsCl, 90 K-Gluconate, 1.8 NaCl, 1.7 MgCl$_2$, 3.5 KCl, 0.05 EGTA, 10 HEPES-CsOH (pH 7.4), 2 MgATP, 0.4 Na$_2$-GTP, 10 Phosphocreatine, 4 QX314-Cl (~300 mOsm). Cells were held at at −70 mV for AMPAR-EPSC and IPSC recordings, at +40 mV for NMDAR-EPSC recordings and. Access resistance was monitored during recording and cells were discarded if access exceeded 15 MΩ or 20 MΩ during recording of evoked or spontaneous synaptic currents, respectively. Action potentials in presynaptic neurons were elicited with a bipolar focal stimulation electrode fabricated from nichrome wire. The RRP in *Figure 4* and *5* was measured by application of 0.5 M sucrose in extracellular solution applied via a microinjector syringe pump for 10 s at a flow rate of 10 µL/min. For *Figure 5—figure supplement 2*, the RRP was measured by focal application of 0.5 M sucrose with a picospritzer for 30 s in the presence of TTX (1 µM), as described in *Liu et al. (2014)*. Data were acquired with an Axon 700B Multiclamp amplifier and digitized with a Digidata 1440A digitizer. For action potential and sucrose-evoked responses, data were acquired at 5 kHz and low-pass filtered at 2 kHz. For miniature recordings data were acquired at 10 kHz. All data acquisition and analysis was done using pClamp10. For all electrophysiological experiments, the experimenter was blind to the genotype throughout data acquisition and analysis.

## Presynaptic Ca$^{2+}$ imaging

All Ca$^{2+}$-imaging experiments were done in cultured hippocampal neurons infected with lentiviruses (expressing active cre or inactive cre) at DIV 5. Presynaptic Ca$^{2+}$ transients were examined at DIV15 - 18 in whole cell patch clamp configuration at room temperature. The extracellular solution contained (in mM): 140 NaCl, 5 KCl, 2 CaCl$_2$, 2 MgCl$_2$, 10 Glucose, 0.05 APV, 0.02 CNQX, 0.05 PTX, 10 HEPES-NaOH (pH 7.4, ~310 mOsm). Glass pipettes were filled with intracellular solution containing (in mM) 140 K Gluconate, 0.1 EGTA, 2 MgCl$_2$, 4 Na$_2$ATP, 1 NaGTP, 0.3 Fluo-5F, 0.03 Alexa Fluor 594, 10 HEPES-KOH (pH 7.4, ~300 mOsm). Neurons were filled for 7 min and axons and dendrites were identified in the red channel. Presynaptic boutons were identified by their typical bead-like morphology. Neurons in which the distinction between axons and dendrites was unclear were discarded. 10 min after break-in, presynaptic Ca$^{2+}$ transients were induced by a single action potential evoked via somatic current injection (5 ms, 800–1200 pA) and monitored via Fluo-5F fluorescence. Images were acquired using an Olympus BX51 microscope with a 60x, 1.0 numerical aperture objective. Fluorescence signals were excited by a light-emitting diode at 470 nm, and were collected with a scientific complementary metal–oxide–semiconductor camera at 100 frames/s for 200 ms before and 1s after the action potential. After the experiment, coverslips were fixed using 4% paraformaldehyde/phosphate-buffered saline. Neurons were stained with GAD67 antibodies as described above and imaged neurons, identified post-hoc by their Alexa 594 filling, were identified as either inhibitory or excitatory neurons using confocal microscopy. In additional experiments (*Figure 3—figure supplement 1*), distinction between excitatory and inhibitory cultured neurons was further characterized by labeling excitatory neurons using a lentivirus expressing tandem dimer tomato (TdTomato)

under a CamKII promoter (pFCK(1.3)tdimer2W; gift from Pavel Osten; Addgene plasmid #27233, [*Dittgen et al., 2004*]). $Ca^{2+}$ transients were quantified using ImageJ. For the analysis in boutons, ROIs were defined using pictures taken in the red channel, and 7–10 boutons were randomly selected from each neuron. Background was subtracted using the rolling ball method with a radius of 1.5 μm. After background subtraction, $(F-F_0)/F_0$ was calculated (F = average green emission in a bouton at a given time point, $F_0$ = average fluorescent intensity in frames 0 to 20 before action potential induction). For dendritic measurements, a second order dendrite was selected from each neuron, and the fluorescence from a nearby empty region was referred to as background and subtracted from the ROI. For all $Ca^{2+}$ imaging experiments, the experimenter was blind to the genotype throughout data acquisition and analysis.

## Statistics

Statistical significance was set at $*p \leq 0.05$, $**p \leq 0.01$, and $***p \leq 0.001$. Unless otherwise noted in the figure legends, all tests were performed using Student's t tests to compare means. In cases where significance was determined by one-way ANOVA, Holm-Sidak multiple comparisons tests were used to compare all conditions against the cDKO condition to assess rescue. All experiments were done with using a minimum of three independent cultures and in each culture multiple cells (typically 5–10 per culture and genotype) or images (typically 5–10 images per culture and genotype) were analyzed.

## Acknowledgements

We thank Lydia Bickford and Jennifer Wang for technical support, Dr. Nils Brose and Dr. Thomas Südhof for antibodies, and Dr. Wade Regehr and all members of the Kaeser laboratory for insightful discussions. This work was supported by grants from the NIH (F31NS089077 to RGH and R01NS083898 to PSK), the Nancy Lurie Marks Foundation (to PSK), the Brain Research Foundation (to PSK), the Lefler Foundation (to PSK), the Alice and Joseph Brooks Fund (to CL) and the Harvard Brain Initiative (to PSK). We acknowledge the Neurobiology Imaging Facility, supported by a NINDS P30 Core Center Grant (NS072030), for instrument availability and consultation.

## Additional information

### Funding

| Funder | Grant reference number | Author |
|---|---|---|
| National Institute of Neurological Disorders and Stroke | R01NS083898 | Pascal S Kaeser |
| Nancy Lurie Marks Family Foundation | Junior Faculty MeRIT fellowship | Pascal S Kaeser |
| Brain Research Foundation | Seed Grant, BRFSG-2013-04 | Pascal S Kaeser |
| Alice and Joseph Brooks Foundation | Postdoctoral Fellowship | Changliang Liu |
| Harvard Brain Initiative | HBI Bipolar Disorder Grant | Pascal S Kaeser |
| National Institute of Neurological Disorders and Stroke | Graduate student fellowship, F31NS089077 | Richard G Held |
| Lefler Foundation | Lefler Small Grant Award | Pascal S Kaeser |

The funders had no role in study design, data collection and interpretation, or the decision to submit the work for publication.

### Author contributions

RGH, PSK, Conception and design, Acquisition of data, Analysis and interpretation of data, Drafting or revising the article; CL, Acquisition of data, Analysis and interpretation of data

## Author ORCIDs

Pascal S Kaeser, http://orcid.org/0000-0002-1558-1958

## Ethics

Animal experimentation: All experiments using mice were performed according to institutional guidelines at Harvard University, and were in strict accordance with the recommendations in the Guide for the Care and Use of Laboratory Animals of the National Institutes of Health. The animals were handled according to protocols (protocol number IS00000049) approved by the institutional animal care and use committee (IACUC).

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
