## [Decision Letter]

Thank you for submitting your article "ELKS controls the pool of readily releasable vesicles at excitatory synapses through its N-terminal coiled-coil domains" for consideration by *eLife*. Your article has been reviewed by three peer reviewers, one of whom is a member of our Board of Reviewing Editors and the evaluation has been overseen by Gary Westbrook as the Senior Editor.

The reviewers have discussed the reviews with one another and the Reviewing Editor has drafted this decision to help you prepare a revised submission.

Summary:

In this paper, the authors study the function of ELKS at excitatory synapses in cultured neurons. Previous work showed that at inhibitory synapses ELKS stimulates release probability (P) by stimulating calcium influx. Here, the authors find that the removal of both ELKS proteins cause a reduction in Readily Releasable Pool (RRP) in excitatory synapses, indicating that the mechanism of action of ELKS proteins at inhibitory and excitatory synapses is fundamentally different. Molecular dissection shows that the three N-terminal, but not the C-terminal, coiled-coils are responsible.

Essential revisions:

All reviewers agree that the study is interesting and novel, but that more evidence is required to unequivocally prove that the molecular function of the ELKS proteins is indeed fundamentally different in excitatory and inhibitory synapses. Most importantly, this requires, in our opinion, that key experiments are carried out in parallel at excitatory and inhibitory neurons instead of basing all conclusions on comparing the present data with the previously published work. In the prior J. Neurosci. paper, the conclusion that the RRP was unchanged was based on a low number of cells (and on average the RRP was lower in the cDKO) – and the finding that increasing calcium concentrations did not rescue release in inhibitory neurons indicates that the RRP must have been lower. Conversely, in the present paper the PPR was actually slightly higher in the cDKO (Figure 3), and comparing the change in AP-evoked and sucrose-evoked EPSC (Figure 1 and Figure 4) seems to indicate that the vesicular release probability is reduced in the cDKO. Moreover, the results with single KOs also indicate that ELKS affect the RRP in inhibitory synapses, as discussed by the authors. Therefore, the question arises whether ELKS has effects on both P and RRP in both types of synapses, but due to a different organization of the two types of terminals, one effect will be more conspicuous in inhibitory synapses, and another effect will be conspicuous in excitatory synapses. In this case, the differences will not be indicative of a different function of ELKS in the two neuronal types. Thus, we recommend parallel experiments of release probability and RRP in excitatory and inhibitory neurons from the same cultures, while matching the powers of the analysis (i.e. similar number of cells in the two types of neurons). Furthermore, the statistical analysis for multiple comparisons is not appropriate and needs to be corrected.

The second major issue is that there is presently no mechanistic/molecular explanation for the difference between the synapses, particularly when considering that the molecular machinery involved in vesicle docking and fusion is very similar. While we understand that addressing this point may exceed the scope of this study, this is puzzling and needs to be thoughtfully discussed.

---

## [Author Response]

Essential revisions:

*All reviewers agree that the study is interesting and novel, but that more evidence is required to unequivocally prove that the molecular function of the ELKS proteins is indeed fundamentally different in excitatory and inhibitory synapses. Most importantly, this requires, in our opinion, that key experiments are carried out in parallel at excitatory and inhibitory neurons instead of basing all conclusions on comparing the present data with the previously published work.*

We thank the reviewers for their positive assessment of our work. As suggested, we have performed a systematic comparison of vesicular release probability and the size of the RRP for excitatory and inhibitory transmission. All eight parameters (PPR and RRP, excitatory and inhibitory synapses, cDKO and control) were measured in each of three cultures and with identical sample numbers. These data are now presented in a new figure (Figure 5). The results of these experiments match with both our previously published experiments from Liu et al. 2014 and the data included in the previous version of the manuscript.

In the prior J. Neurosci. paper, the conclusion that the RRP was unchanged was based on a low number of cells (and on average the RRP was lower in the cDKO) – and the finding that increasing calcium concentrations did not rescue release in inhibitory neurons indicates that the RRP must have been lower.

As pointed out by the reviewers, the RRP measured at inhibitory synapses in the Liu et al. 2014 paper revealed a very slight trend toward a reduction, though this difference was not significant. Similarly, the inhibitory RRP measured in our current experiments, using a larger sample number, is slightly below the control but the difference is not significant (p = 0.2440). Furthermore, we have included a third independent experiment (Figure 5—figure supplement 2), which again failed to detect a significant difference in RRP size at inhibitory synapses. However, as we have pointed out in the Discussion there is clearly some role for ELKS in RRP at inhibitory synapses that our methods cannot easily capture. This is evident in the Ca^2+^ titration experiments shown in Liu et al. 2014 and in the ELKS2α single cKO analyzed in Kaeser et al. 2009. While we acknowledge and discuss these findings, we do not feel that they negate the main observation that removal of ELKS1α/2α has differential effects on the release properties of different synapses.

Conversely, in the present paper the PPR was actually slightly higher in the cDKO (Figure 3), and comparing the change in AP-evoked and sucrose-evoked EPSC (Figure 1 and Figure 4) seems to indicate that the vesicular release probability is reduced in the cDKO.

In our new experiment in Figure 5 the slight difference pointed out by the reviewers in Figure 3 appears to trend in the opposite direction. We believe that these differences simply reflect the variation inherent in the experiment, as opposed to a true change in vesicular release probability. Furthermore, at inhibitory synapses the reduction in P is caused by reduced Ca^2+^-influx, which is unaffected at excitatory synapses. Similarly, as discussed in detail below in response to reviewer 2, division of the AP-evoked charge transfer by the sucrose evoked RRP charge requires that both parameters be measured from the same inputs and the same cells. The data presented in Figure 1 and Figure 4 were collected from independent cultures and we hesitate to attribute differences in phenotype magnitude in the range of 10% to anything other than variability between cultures.

Moreover, the results with single KOs also indicate that ELKS affect the RRP in inhibitory synapses, as discussed by the authors. Therefore, the question arises whether ELKS has effects on both P and RRP in both types of synapses, but due to a different organization of the two types of terminals, one effect will be more conspicuous in inhibitory synapses, and another effect will be conspicuous in excitatory synapses. In this case, the differences will not be indicative of a different function of ELKS in the two neuronal types. Thus, we recommend parallel experiments of release probability and RRP in excitatory and inhibitory neurons from the same cultures, while matching the powers of the analysis (i.e. similar number of cells in the two types of neurons).

We thank the reviewers for the suggestion of this systematic comparison and performed this exact experiment. Comparing release probability and RRP in excitatory and inhibitory neurons from the same cultures (with matched sample size) confirms our initial finding of a significant reduction of RRP (with normal PPR) at excitatory synapses, and a significant reduction of P (with near normal RRP) at inhibitory synapses. The results are presented in the new Figure 5.

In respect to the interpretation of this result, we did not intend to make the point that ELKS serves completely separable, non-overlapping molecular functions at inhibitory versus excitatory synapses. Our own previous work has demonstrated that ELKS can regulate the RRP at inhibitory synapses and we cannot entirely rule out small effects on release probability at excitatory synapses in the ELKS cDKO. We can, however, confidently state that removal of ELKS1α/2α has differential effects on the properties of release (i.e. RRP size and release probability) at different synapses. This observation is significant because it is clear that synaptic properties between inhibitory and excitatory synapses are different, but molecular differences at these synapses are not well understood. ELKS removal has differential effects on transmission and thus strongly suggests that some element of the active zone, be it its molecular components or the structural or functional organization of the same components, are different between synaptic subtypes. Our study is one of the first that starts dissecting such differences by manipulations at the molecular level.

We apologize that we did not make this more clear in the initial version, and we have made changes throughout the text to soften the impression that our findings imply a hard binary function of ELKS at inhibitory and excitatory synapses.

Furthermore, the statistical analysis for multiple comparisons is not appropriate and needs to be corrected.

We thank the reviewers for pointing out this error and we have changed our analysis to compare each condition against the cDKO and now use one-way ANOVA followed by a Holm-Sidak post-hoc test, which corrects for multiple comparisons. We have also added additional rescue data to the new Figure 7 and Figure 8 to strengthen our conclusions.

The second major issue is that there is presently no mechanistic/molecular explanation for the difference between the synapses, particularly when considering that the molecular machinery involved in vesicle docking and fusion is very similar. While we understand that addressing this point may exceed the scope of this study, this is puzzling and needs to be thoughtfully discussed.

We agree with the reviewers that this is a puzzling point, but we also think it is important. A large body of literature supports that synapses are functionally diverse, but the molecular diversity that underlies this functional diversity is not well understood. Even though differences in the molecular make up of release sites between excitatory and inhibitory synapses are currently not well known, the molecular mechanisms behind the functional differences are important to resolve. It is interesting that a recent paper (Boyken et al., 2013) found few differences between docking sites of inhibitory vs. excitatory synapses, but there were differences in their levels of Bassoon, ELKS1, and ELKS2 (see supplemental table 5, Boyken et al., 2013). At this time, no single potential mechanism stands out in our data, and no previous study has found a compelling molecular explanation for the functional differences. We have made an effort throughout the manuscript to thoughtfully address potential mechanisms in experiment (using structure-function rescue, Figure 7 and Figure 8) and text. In the revised manuscript, we have added new data to show that the sequences that are necessary to rescue the excitatory RRP are not needed for rescue of the IPSC, further supporting differential functions of ELKS at different synapses. In the Discussion, we have outlined which protein interactions map on these sequences, which gives first hints towards mechanisms.

Unambiguously determining the exact molecular mechanism is difficult at this point and we strongly feel that it goes beyond the scope of this study. In order to directly test each individual protein interaction without affecting other functions/interactions of ELKS, exact structural information on the binding sites would be necessary. Unfortunately, structural information on ELKS and its interactions with potential targets such as Liprin-α or Bassoon is not available. It is thus not easily possible to dissect roles for potential interactions with precise mutations in the α-helical coils of ELKS that only affect one interaction at a time. Identification of point mutations to selectively change individual interactions would require a massive screening or structural effort, which is not possible for a revision.

We hope that the reviewers agree that with the new data, the additional considerations in Results and Discussion, and in view of these explanations, our structure-function analysis provides significant advance, but that an exact molecular understanding of ELKS function at each synapse will require much more groundwork on the structure of these fascinating but enigmatic proteins.